# Impaired Organokine Regulation in Non-Diabetic Obese Subjects: Halfway to the Cardiometabolic Danger Zone

**DOI:** 10.3390/ijms24044115

**Published:** 2023-02-18

**Authors:** Hajnalka Lőrincz, Balázs Ratku, Sára Csiha, Ildikó Seres, Zoltán Szabó, György Paragh, Mariann Harangi, Sándor Somodi

**Affiliations:** 1Division of Metabolism, Department of Internal Medicine, Faculty of Medicine, University of Debrecen, H-4032 Debrecen, Hungary; 2Department of Emergency Medicine, Faculty of Medicine, University of Debrecen, H-4032 Debrecen, Hungary; 3Doctoral School of Health Sciences, University of Debrecen, H-4032 Debrecen, Hungary; 4Institute of Health Studies, Faculty of Health Sciences, University of Debrecen, H-4032 Debrecen, Hungary

**Keywords:** obesity, type 2 diabetes, insulin resistance, afamin, plasminogen-activator inhibitor-1, retinol binding protein 4, lipid metabolism, lipoprotein subfraction, cardiometabolic

## Abstract

Altered organokine expression contributes to increased cardiometabolic risk in obesity. Our aim was to evaluate the associations of serum afamin with glucose homeostasis, atherogenic dyslipidemia, and other adipokines in severe obesity to clarify the early metabolic alterations. 106 non-diabetic obese (NDO) subjects and 62 obese patients with type 2 diabetes matched for age, gender, and body mass index (BMI) were enrolled in this study. We compared their data with 49 healthy, lean controls. Serum afamin and retinol-binding protein 4 (RBP4), as well as plasma plasminogen activator inhibitor-1 (PAI-1), were measured with ELISA, and lipoprotein subfractions were analyzed using Lipoprint gel electrophoresis. Afamin and PAI-1 found to be significantly higher in the NDO and T2M group (*p* < 0.001 and *p* < 0.001, respectively) than in the controls. In contrast, RBP4 was unexpectedly lower in the NDO and T2DM group compared to controls (*p* < 0.001). Afamin showed negative correlations with mean LDL size and RBP4, but positive correlations with anthropometric, glucose/lipid parameters, and PAI-1 in both the overall patients and the in NDO + T2DM groups. BMI, glucose, intermediate HDL, and small HDL were predictors of afamin. Afamin may serve as a biomarker for the severity of cardiometabolic disturbances in obesity. The complexity of organokine patterns in NDO subjects draws attention to the diverse spectrum of obesity-related comorbidities.

## 1. Introduction

Obesity is generally regarded as a condition with an adverse cardiometabolic risk profile; however, several studies have suggested that a large proportion of obese individuals do not have relevant cardiometabolic risk factors [1]. Some studies demonstrated that obese patients without metabolic abnormalities are not at increased risk for type 2 diabetes (T2DM) and cardiovascular disease (CVD) [2,3,4], while others, including large meta-analyses, provided evidence indicating a higher risk for CVD events even in metabolically healthy obese (MHO) patients [5]. The former concept that normal metabolism exists in any obesity phenotype has been highly debated [6]. Indeed, it is more plausible that obese patients demonstrate metabolic abnormalities, but these alterations are not necessarily detectable using classical criteria for metabolic syndrome (MetS) [7]. Therefore, the focus of studies have shifted towards evaluating organokines, including hepatokines, adipokines, and gut hormones, which might reveal subtle changes in metabolic homeostasis and might also reflect the severity of cardiometabolic disturbance.

The vitamin E binding afamin is considered one of the hepatokines that may play a key role in the pathogenesis of obesity-related diseases. In large epidemiological studies, afamin was strongly associated with the prevalence of MetS and was correlated with each component of MetS [8]. In another multicenter study, afamin levels were found to be related to prediabetes, insulin resistance, and the prevalence of T2DM. In addition, afamin was suggested to be a potential biomarker for identifying individuals at high risk of T2DM [9]. Importantly, Shen et al. demonstrated a direct effect of afamin on the key enzymes of glucose metabolism in vitro, indicating a causal link between afamin and T2DM [10]. More recently, increased afamin concentrations were also observed in nonalcoholic fatty liver disease (NAFLD) [11,12].

Plasminogen activator inhibitor-1 (PAI-1) is an endogenous regulator of fibrinolysis and a well-known adipokine with a putative role in the development of obesity-related diseases [13,14]. Similar to afamin, PAI-1 levels are strongly correlated with the components of MetS and effectively predict T2DM [15]. PAI-1 has also been suggested to have a causal effect on the risk of coronary heart disease, potentially mediated through the dysregulation of glucose metabolism [16].

Retinol binding protein-4 (RBP4) is a transport protein for retinol and a controversial organokine, which is also suggested to link obesity with its complications, particularly insulin resistance, T2DM, and MetS [17,18,19]. Elevated RBP4 levels were associated with increased cardiovascular risk and adverse cardiovascular events in patients with chronic heart failure [20]. 

Non-diabetic obese (NDO) individuals can be characterized by metabolic disturbances, including atherogenic dyslipidemia, and early changes in organokine expression, but their glucose metabolism is still within normal ranges. Our research group has recently demonstrated higher afamin levels in a small group of NDO individuals compared to lean subjects [21]. In the present study, our primary objective was to challenge the concept of unaffected metabolism in obesity and demonstrate evidence for cardiometabolic disturbance in NDO subjects. Therefore, we measured afamin concentrations in a larger group of NDO and a group of obese patients with T2DM and evaluated the relationship of afamin with parameters of lipid- and glucose metabolism. Because PAI-1 and RBP4 have shown similar pattern of changes in obesity-related diseases, but data regarding their relationship with afamin is almost entirely lacking, the associations of afamin with PAI-1 and RBP4 were also investigated. We hypothesized that carbohydrate and lipid parameters, including lipoprotein subfractions, correlate with circulating afamin in both study groups. Additionally, we expect significantly higher organokine levels (afamin, PAI-1, and RBP4) in severe obese T2DM patients.

## 2. Results

Anthropometric and routine laboratory parameters, as well as major medications, are summarized in Table 1. Compared to controls, both the NDO group and the T2DM group had significantly higher BMI, waist circumference, hsCRP, fasting glucose, fasting insulin, ALT, γ-GTP, and triglyceride levels. In addition, fasting glucose, sTSH, and triglyceride levels were significantly higher in patients with T2DM compared to NDO subjects, while HDL-C was significantly lower in patients with T2DM compared to lean controls. The mean serum afamin concentration was found to be 32.2% higher in the NDO group and nearly two-fold higher in the T2DM group compared to normal weight ones (controls: 56 ± 30.3 vs. NDO: 82.6 ± 19.7 μg/mL vs. T2DM: 109.2 ± 21.4 μg/mL, ANOVA: *p* < 0.001) (Table 1). In all patients overall and NDO subjects, afamin was significantly higher among men (all: 91.9 ± 30.3 μg/mL (male) vs. 83.4 ± 24.5 μg/mL (female), *p* = 0.03; and NDO: 91 ± 20.8 μg/mL (male) vs. 80.2 ± 18.8 μg/mL (female), *p* = 0.02).

Analyzing the distribution of LDL subfractions, the percentages of VLDL, large LDL, and small-dense LDL subfractions were significantly higher in NDO and T2DM compared to controls (Table 2). In terms of the absolute number of subfractions, these tendencies were similar. The mean LDL particle size was found to be significantly lower in both NDO and T2DM subjects compared to lean ones. In line with literature data, there was a shift towards small-sized HDL subfractions in NDO and T2DM patients: the percentage and amount of large HDL subfractions were significantly lower, while the percentage of small HDL subfractions was significantly higher in these groups (Table 2 and Figure 1). 

Serum afamin showed significant positive correlations with age (r = 0.17; *p* = 0.01), BMI (r = 0.39; *p* < 0.001), and waist circumference (r = 0.55; *p* < 0.001) in overall participants. Furthermore, significant positive correlations were found between fasting glucose, HbA1C, fasting insulin, and afamin (Figure 2a–c). 

The percentage of IDL subfractions and mean LDL size correlated negatively with serum afamin (r = −0.29; <0.001 and r = −0.29; *p* < 0.001, respectively); while the percentage of large LDL and small-dense LDL subfractions correlated positively with afamin (r = 0.38; *p* < 0.001 and r = 0.19; *p* < 0.01; respectively) in overall participants. The percentage of VLDL subfraction did not correlate with afamin. 

Among HDL subfractions, HDL-1 to -5 subfractions, which correspond with large HDL and partially with intermediate HDL, showed strong, significant negative correlations with serum afamin in all subjects (Table 3 and Figure 2d,e). While there were strong positive correlations between serum afamin and HDL-7 to -10 subfractions, which correspond partially with intermediate and mainly with small HDL subfractions in overall subjects (Table 3 and Figure 2f). 

Circulating RBP4 was significantly lower in the NDO and T2DM groups than normal weight individuals (controls: 41.4 ± 14.4 μg/mL vs. NDO: 32.3 ± 15 μg/mL vs. T2DM: 28.8 ± 12.3 μg/mL; one-way ANOVA: *p* < 0.001) (Figure 3a). There was a negative correlation between RBP4 and afamin (r = −0.22; *p* = 0.004) (Figure 3b). Plasma PAI-1 was significantly higher in the NDO and T2DM groups (controls: 3.63 (1.99–7.29) ng/mL vs. NDO: 7.37 (4.94–10.42) ng/mL vs. T2DM: 6.62 (4.6–10.28) ng/mL; Kruskal-Wallis H test: *p* < 0.001) (Figure 3c) and a significant positive correlation can be seen between plasma PAI-1 and afamin in overall subjects (r = 0.21; *p* = 0.002).

All correlations were observed in both the overall study population (n = 217) and the NDO + T2DM groups (n = 168). Since several correlations were observed between afamin and anthropometric/laboratory parameters, a backward stepwise multiple regression analysis was performed to determine the significant predictor(s) of afamin. The model included gender, age, BMI, mean LDL size, fasting glucose, large HDL (mmol/L), intermediate HDL (mmol/L), and small HDL (mmol/L). The analysis showed that BMI (β = 0.214; *p* < 0.001), fasting glucose (β = 0.291; *p* < 0.001), intermediate HDL (mmol/L) (β = −0.36; *p* < 0.001), and small HDL (mmol/L) (β = 0.446; *p* < 0.001) were independent predictors of afamin.

## 3. Discussion

It has been widely established that atherogenic dyslipidemia is a significant contributor to the increased cardiovascular risk in obese and T2DM patients. High levels of triglyceride-rich lipoproteins, including VLDL and IDL particles, as well as a high percentage of small-dense LDL, have been found to be associated with insulin resistance in non-obese patients with impaired fasting glucose and T2DM [22]. Moreover, a genome-wide association study suggested that obese women may be more susceptible to aggregated genetic risk related to triglyceride-raising alleles; in addition, higher BMI significantly correlated with higher concentrations of very large, large, medium, and small triglyceride-rich lipoprotein subfractions measured by magnetic resonance spectroscopy [23]. The findings of the present study support the presence of an atherogenic lipid profile in obesity. Compared to lean controls, we detected higher triglyceride, VLDL, large LDL, small-dense LDL levels, and a smaller mean LDL size in NDO and obese T2DM patients. In addition, we observed a shift towards small-sized HDL subfractions in the NDO and obese T2DM groups compared to lean controls. The latter finding is in accordance with a recent observation, which reported the accumulation of small HDL particles in metabolically unhealthy overweight and obese subjects [24]. Furthermore, in a large Chinese cohort, lower large HDL-C and higher small HDL-C were independently related to the presence of T2DM [25]. It must be noted that measurement of conventional lipid parameters, i.e., HDL-C, is not entirely suitable for predicting the risk of CVD in diabetes; therefore, studying lipoprotein subfractions is of particular importance. As a matter of fact, HDL-C concentrations do not necessarily reflect the concentration of HDL particles and the important anti-atherogenic properties of HDL. Thus, due to the complex nature of HDL, the recent focus has shifted from HDL quantity to HDL quality [26,27]. 

Former studies have reported a strong association of elevated serum afamin with various metabolic disorders, including obesity, MetS, T2DM, NAFLD, and coronary atherosclerosis [9,28,29]. In line with these data, in our study, afamin levels were found to be 32.2% higher in the NDO group and nearly two-fold higher in the T2DM group, suggesting that elevated afamin may indicate metabolic disturbance in severe obesity with intact glucose parameters. Our results underline the usefulness of measuring serum afamin levels in severe obesity, which demonstrate early alterations in the regulation of liver-derived hormone-like peptides. 

Afamin showed positive associations with anthropometric parameters, systolic blood pressure, liver enzymes, glucose parameters, and obesity-related dyslipidemia, suggesting a potentially harmful effect of this hepatokine on cardiometabolic processes [8,30]. In our study, afamin correlated positively with BMI and waist circumference and showed strong positive correlations with carbohydrate parameters, including fasting glucose, fasting insulin, and HbA1C, in overall subjects and in the NDO and T2DM groups. Moreover, afamin showed significant associations with an altered lipid profile, such as mean LDL size as well as large and small-dense LDL subfractions. Interestingly, triglyceride-rich lipoproteins, i.e., VLDL did not correlate with afamin. During Lipoprint electrophoresis, bidirectional associations were detected between HDL subfractions and afamin in overall subjects and in the NDO and T2DM groups. The large-sized HDL subfractions (HDL1–5) correlated negatively, while the small-sized HDL subfractions (HDL7–10) correlated positively with afamin. Multiple regression analysis determined that in addition to BMI and fasting glucose, the levels of intermediate and small HDL were also significant predictors of afamin. Our observations were similar to those published by Jerkovic et al., demonstrating that circulating afamin was partially coeluted with apoAI containing small-sized HDL subfractions measured by size-exclusion chromatography [31]. 

A growing number of investigations have described the beneficial and harmful roles of organokines in the pathogenesis of obesity. Based on these findings, it seems that it is not sufficient to study these bioactive hormones separately, but due to the dynamic interplay of their autocrine, paracrine, and endocrine actions, they require a complex examination. Apart from their effects on the endocrine system, in obesity, organokines have a fine-tuning role in the regulation of metabolic processes as well as in the pathogenesis of atherosclerosis [32]. For instance, besides its regulatory function in fibrinolysis, PAI-1 plays an important role in obesity, MetS, diabetic complications, systemic inflammation, and tumor progression, which highlights the various pathophysiological functions of this serpin protein [33]. In agreement with previous results, we showed significantly higher plasma PAI-1 levels in NDO and T2DM subjects compared to lean ones [34,35,36]. Currently only preliminary findings on the interplay of organokine regulation are available. To the best of our knowledge, this is the first report to demonstrate a strong correlation between plasma PAI-1 and afamin levels. 

Urinary and serum proteomic analyses showed that RBP4 and afamin were upregulated and positively correlated in children with NAFLD [37]. Furthermore, measurement of urinary afamin and RBP4 may serve as biomarkers for the accurate diagnosis of diabetic nephropathy and might help to reduce the burden associated with renal biopsy in patients with diabetic nephropathy [38]. These results also emphasize the importance of the complex evaluation of organokines in obese patients. 

Despite previous epidemiological observations, we found unexpectedly lower RBP4 both in NDO and T2DM groups compared to lean controls. These data are in line with data from a previous review demonstrating inconclusive associations with the role of RBP4 in the pathogenesis of T2DM [39]. Indeed, high levels of RBP4 may induce insulin resistance via its effects on glucose clearance because of the potentially altered adipose tissue regulation [40]. A U-shaped association has been described between RBP4 level and the incidence of T2DM [41,42]; however, Schiborn et al. mentioned that this association was detected only in women [42]. A Thai study showed that high circulating RBP4 was a predictor of insulin resistance and the severity of coronary artery disease (CAD) in T2DM with CAD [43]. However, it must be noted that Asian race, lower BMI (~25 kg/m^2^), severe comorbidities, i.e., CAD, and intensive insulin therapy can shade these results and make it difficult to compare them with our data. Single nucleotide polymorphisms (SNPs) in the promoter region of the *RBP4* gene may also be associated with an elevated serum RBP4 level, and different RBP4 phenotypes can be related to carbohydrate metabolism in patients with T2DM [44]. Further studies are needed to clarify this phenomenon.

Limitations of this study are the small sample of T2DM patients and the relatively small portion of male patients, which may also reduce the power of the study. However, our results draw attention to the importance of studying organokine metabolism in a complex way in severe obesity. Determination of SNPs in the *RBP4* gene may add further valuable information about the genetic background of diabetic complications. A long-term follow-up—which also investigates the future cardiovascular outcomes, i.e., CAD, CVD, and stroke—may clarify the role of organokines in the development of obesity-related cardiometabolic complications.

## 4. Materials and Methods

### 4.1. Patient Enrollment

106 NDO subjects and 62 obese patients with T2DM were enrolled from our obesity and diabetes outpatient clinics at the Department of Internal Medicine, Faculty of Medicine, University of Debrecen, Hungary. 49 healthy, lean volunteers were also enrolled as controls in our study. All three groups were matched for gender and age. Obesity was defined as a body mass index (BMI) ≥30 kg/m^2^. Patients with endocrine, liver, kidney, pulmonary, neurological, gastrointestinal, acute infection, autoimmune disease, or malignancies were excluded. Further exclusion criteria were pregnancy, lactation, smoking, and regular alcohol consumption. Besides anthropometric and laboratory data, we also assessed the medications of the enrolled subjects. Patients with T2DM were treated either with antidiabetics (mostly metformin and glucagon-like peptide-1 receptor agonists) or with insulin. The participants were referred to scheduled medical appointments from 08:00–10:00 am, and we asked the patients to arrive after an overnight fast. All participants provided written, informed consent. Permission to carry out this study was granted by the Regional Ethics Committee of the University of Debrecen and the Medical Research Council (registration numbers: DE RKEB/IKEB 5513B-2020 and IV/7989-1/2020/EKU, respectively).

### 4.2. Sample Collection

After an overnight fast, venous blood samples will be collected into Vacutainer^®^ tubes (Becton Dickinson, San Jose, CA, USA). Sera and EDTA-anticoagulated plasmas were separated at 3500 g for 15 min 4 °C. Routine laboratory parameters (high sensitivity C-reactive protein [hsCRP], fasting glucose, hemoglobin A1C [HbA1C], glomerular filtration rate [GFR], liver enzymes, supersensitive thyroid stimulating hormone [sTSH], triglyceride, total cholesterol, low-density lipoprotein-cholesterol [LDL-C], high-density lipoprotein-cholesterol [HDL-C]) were performed with a Cobas c600 autoanalyzer (Roche Ltd., Mannheim, Germany) at the Department of Laboratory Medicine, Faculty of Medicine, University of Debrecen, Hungary. Reagents were purchased from the same vendor, and tests were performed according to the recommendations of the manufacturer. Samples were kept at −70 °C before subsequent measurements.

### 4.3. Measurement of Afamin

Serum afamin concentrations were measured by a commercially available enzyme-linked immunosorbent assay (ELISA) kit (Afamin Human ELISA, cat. number: RD194428100R, BioVendor, Brno, Czech Republic), according to the recommendations of the manufacturer. The intra- and inter-assay variation coefficients were <3.61% and <3.4%, respectively. Samples were used in a 100-fold dilution. 

### 4.4. Measurement of RBP4 and PAI-1

Serum RBP4 level was determined by ELISA (Human RBP4 Quantikine ELISA Kit, cat. number: DRB400, R&D Systems, Abingdon, UK) with 5.7–8.1% intra-assay and 5.8–8.6% inter-assay coefficients according to the manufacturer’s instructions. Samples were used in a 1000-fold dilution. Human plasma serpin E1/PAI-1 level was measured by a commercially available DuoSet ELISA (cat. number: DY1786, R&D Systems, Abingdon, UK). Undiluted samples were used for the PAI-1 measurement. 

### 4.5. LDL Subfraction Analysis 

Up to seven LDL subfractions were distributed based on their size using the Lipoprint System (Quantimetrix Corporation, Redondo Beach, CA, USA) according to the manufacturer’s instructions. 25 µL of the sample were mixed in polyacrylamide gel tubes with 200 µL of Sudan Black containing a loading gel. Tubes were photopolymerized for 30 min and then electrophorized at 3 mA/tubes for 64 min. Each electrophoresis chamber involved a quality control provided by Quantimetrix (Liposure Serum Lipoprotein Control, Quantimetrix Corp., Redondo Beach, CA, USA). Subfraction bands were scanned with an ArtixScan M1 digital scanner (Microtek International Inc., CA, USA) and analyzed with the Lipoware Software (Quantimetrix Corp., Redondo Beach, CA, USA). After the VLDL peak, the percentage of midbands C through A mainly corresponded to the intermediate-density lipoprotein (IDL) subfraction in the densitogram. The percentage of large LDL (large LDL %) was defined as the summed percentages of LDL1 and LDL2, whereas the percentage of small LDL (small-dense LDL %) was defined as the sum of LDL3–LDL7. Cholesterol concentrations of LDL subfractions were determined by multiplying the relative area under the curve (AUC) of subfractions by the total cholesterol concentration. The intra-assay precisions were 0.58–7.28% for VLDL, 3.85–11.14% for midbands and 1.05–1.52% for LDL, respectively. The inter-assay precisions were 7.12–9.40% for VLDL, 7.47–10.90% for midbands and 1.26–1.57% for LDL, respectively.

### 4.6. HDL Subfraction Analysis

Up to ten HDL subfractions were distributed based on their size using the Lipoprint System (Quantimetrix Corp., Redondo Beach, CA, USA) according to the manufacturer’s instructions. Briefly, 25 µL of the sample were mixed inpolyacrylamide gel tubes with 300 µL of Sudan Black containing a loading gel. Tubes were photopolymerized for 30 min and then electrophorized at 3 mA/tubes for 54 min. The remaining steps of assay were identical to those of the LDL subfraction test. During HDL subfraction analyses, large (HDL-1 to HDL-3), intermediate (HDL-4 to HDL-7), and small (HDL-8 to HDL-10) HDL subfractions were distributed between VLDL + LDL, and albumin peaks. The cholesterol content of HDL subfractions was calculated by multiplying the HDL-C of the sample. The intra- and inter-assay precisions were 0.90–1.47% and 2.49–4.75%, respectively.

### 4.7. Statistical Methods

Statistical analyses were performed using the Statistica 13.5.0.17 software (TIBCO Software Inc., Tulsa, OK, USA). Graphs were made using Statistica 13.5.0.17 and GraphPad Prism 6.01 (GraphPad Prism Software Inc., San Diego, CA, USA). The relationship between the female/male ratio was calculated with the Chi-square test. The normality of continuous data was tested with the Kolmogorov–Smirnov test. Data were expressed as means ± SD or medians (interquartile range). Comparisons between the control, NDO, and T2DM groups were performed with one-way ANOVAs (Tukey post-hoc test) in the case of normally distributed variables and with a Kruskal-Wallis H test in the case of variables with a non-normal distribution. Pearson’s correlation was used to investigate the relationship between selected variables. Since the distribution of some variables of interest became normal upon base-10 logarithm transformation, we used the log values for correlation analyses in the case of these variables. Multiple regression analyses were performed to determine which variables best predicted afamin concentrations. A *p* ≤ 0.05 as considered statistically significant.

## 5. Conclusions

We aimed to study correlations of afamin with carbohydrate parameters including fasting glucose, HbA1C, and fasting insulin as well as HDL subfractions in severely obese patients with or without T2DM. This is the first study to demonstrate a significant negative correlation between RBP4 and afamin and a strong positive correlation between PAI-1 and afamin levels. Our results may underline the significance of serum afamin measurement in severe obesity. BMI, fasting glucose, and intermediate and small HDL subfractions are found to be predictors of afamin in this study; therefore, afamin may serve as a potential biomarker for the severity of cardiometabolic disturbances, including impaired lipid and/or glucose metabolism in obesity. An altered organokine pattern in NDO subjects draws attention to the existence of different obese phenotypes with a diverse spectrum of obesity-related comorbidities. A complex investigation of organokine pattern may help to develop new therapeutic targets and find the appropriate therapeutic responses in this obese subtype.

## Figures and Tables

**Figure 1 ijms-24-04115-f001:**
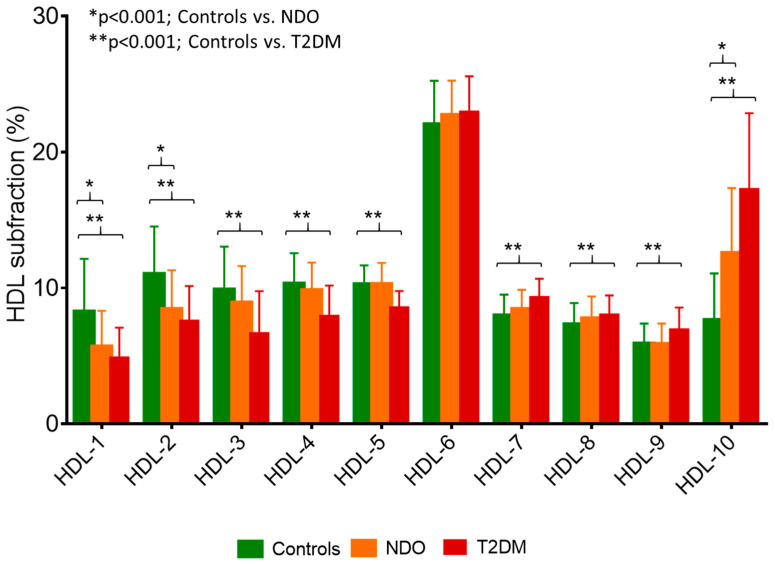
Percentage of high-density lipoprotein (HDL) subfractions in healthy lean subjects (controls; green bar; n = 49), non-diabetic obese patients (NDO; orange bar; n = 106), and patients with type 2 diabetes mellitus (T2DM; red bar; n = 62). Data are presented as mean ± standard deviation. Differences between the three study groups are analyzed using a one-way ANOVA. * indicates a *p* < 0.001 between controls and NDO; ** indicates a *p* < 0.001 between controls and T2DM. HDL subfractions were distributed by Lipoprint gel electrophoresis (Quantimetrix Corp. Redondo Beach, CA, USA) from peripheral blood serum samples.

**Figure 2 ijms-24-04115-f002:**
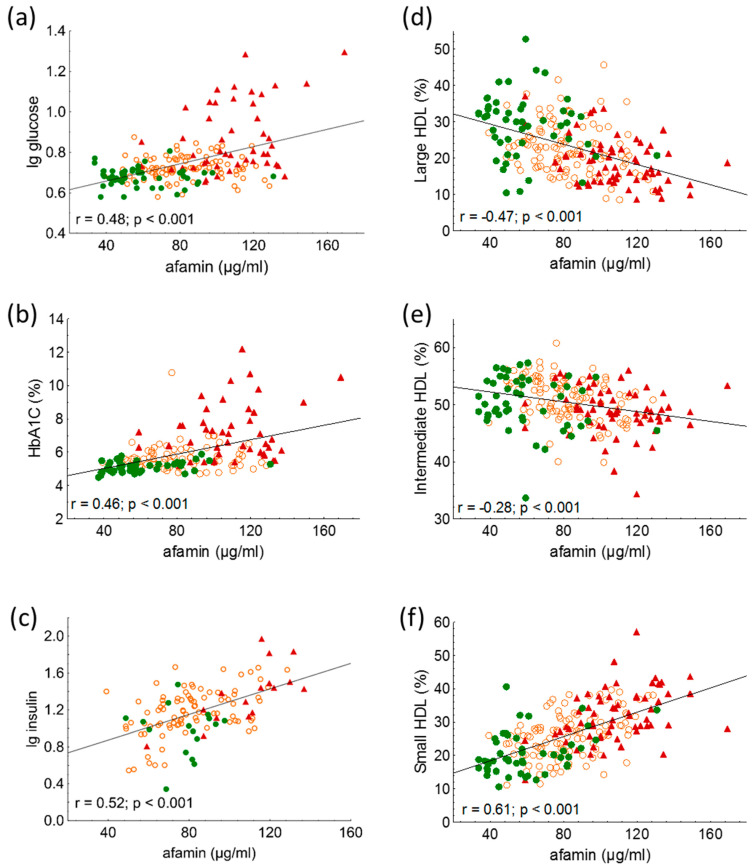
Correlations between (**a**) lg glucose; (**b**) HbA1C; (**c**) lg insulin; (**d**) percentage of large high-density lipoprotein (HDL) subfraction (HDL-1-3); (**e**) percentage of intermediate HDL subfraction (HDL-4-7); and (**f**) percentage of small HDL subfraction (HDL-8-10) and serum afamin in healthy lean subjects (controls; green color), non-diabetic obese patients (NDO; orange color), and patients with type 2 diabetes mellitus (T2DM; red color). Afamin concentrations were quantified from peripheral blood serum samples with enzyme linked immunosorbent assays, and HDL subfractions were distributed by Lipoprint gel electrophoresis.

**Figure 3 ijms-24-04115-f003:**
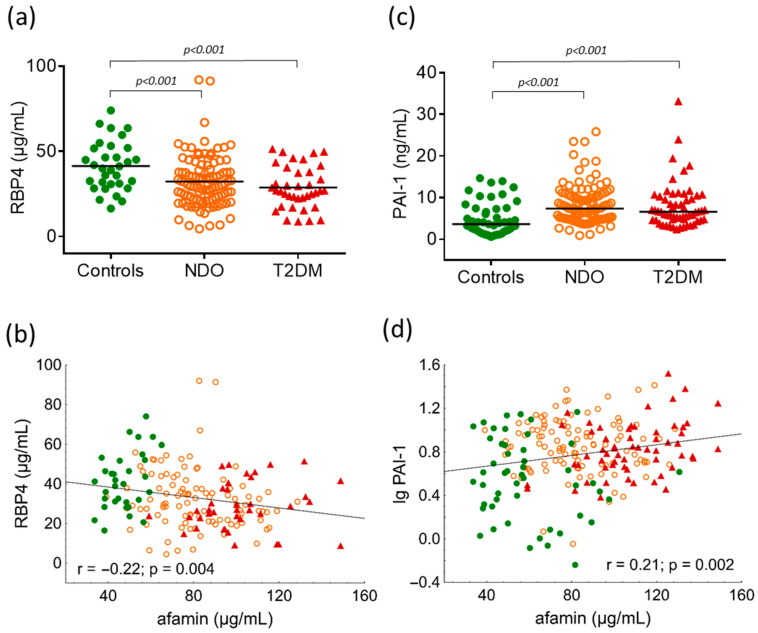
(**a**) Serum levels of retinol binding protein 4 (RBP4) in study groups; (**b**) correlation between RBP4 and afamin in overall subjects; (**c**) plasma levels of plasminogen activator inhibitor-1 (PAI-1) in study groups; and (**d**) correlation between PAI-1 and afamin in overall subjects. Differences between study groups are analyzed using one-way ANOVA (RBP4) and the Kruskal-Wallis H test (PAI-1), respectively. Green dots represent healthy lean subjects (n = 49), orange dots represent non-diabetic obese patients (NDO; n = 106), and red triangles represent patients with type 2 diabetes mellitus (T2DM; n = 62). Afamin and RBP4 concentrations were quantified from peripheral blood serum samples, whereas PAI-1 levels were measured from peripheral blood EDTA-coagulated plasmas with commercially available enzyme-linked immunoassays.

**Table 1 ijms-24-04115-t001:** Anthropometric data, laboratory parameters, and main medications of study participants.

	Controls (n = 49)	NDO (n = 106)	T2DM (n = 62)
**Anthropometric parameters**			
Male/Female (n)	13/36	23/83	22/40
Age (yrs)	43.2 ± 9.1	44.3 ± 12.5	47.6 ± 7.7
BMI (kg/m^2^)	24.7 ± 2.8	42.6 ± 8.1 *	43.1 ± 9.1 ^§^
Waist circumference (cm)	85.2 ± 12.3	123.7 ± 17.4 *	128.3 ± 18.5 ^§^
**Medication**			
Metformin (n, %)	0; 0	11; 10.4	45; 72.8
Insulin (n, %)	0; 0	0; 0	14; 35.5
GLP-1 RA (n, %)	0; 0	0; 0	14; 35.5
Statin (n, %)	0; 0	12; 11.3	26; 41.9
ACEI/ARB (n, %)	1; 2	41; 38.7	28; 45.2
CCB (n, %)	1; 2	15; 14.2	14; 22.6
Diuretics (n, %)	0; 0	22; 20.8	8; 12.9
**Laboratory parameters**			
**afamin (μg/mL)**	**56 ± 20.3**	**82.6 ± 19.7** *	**109.2 ± 21.4** ^§,#^
hsCRP (mg/L)	1.3 (0.6–2.5)	8 (3.4–15.7) *	6.8 (3.1–13.7) ^§^
Glucose (mmol/L)	4.8 (4.5–5.1)	5.2 (4.9–5.8) *	6.4 (5.5–10.5) ^§,#^
HbA1C (%)	5.1 ± 0.3	5.7 ± 0.8 *	7.2 ± 1.7 ^§,#^
Insulin (mU/L)	10.9 (6.6–12.9) (n = 16)	15 (11.2–21.6) *	25.4 (14.1–31.5) (n = 16) ^§^
GFR (mL/1.73 m^2^)	90 (90–90)	90 (90–90)	90 (90–90)
AST (U/L)	19 (17–24)	20 (17–27)	25 (17–30)
ALT (U/L)	17.5 (13–25)	26 (18–35) *	28 (21–44) ^§^
γ-GTP (U/L)	19 (16–28)	28.5 (19–44) *	35 (25–53) ^§^
sTSH (mU/L)	1.65 (1.18–2.11)	1.95 (1.46–2.67)	2.29 (1.34–15.2) (n = 33) ^#^
Triglyceride (mmol/L)	1.1 (0.9–1.5)	1.45 (1.1–1.9) *	1.7 (1.2–2.7) ^§,#^
Total cholesterol (mmol/L)	5 ± 0.8	5 ± 0.8	5 ± 1.2
HDL-C (mmol/L)	1.5 ± 0.4	1.3 ± 0.3	1.2 ± 0.3 ^§^
LDL-C (mmol/L)	2.9 ± 0.5	3.2 ± 0.7	3 ± 0.9

Abbreviations: ACEI/ARB, angiotensin-converting enzyme inhibitors/angiotensin II receptor blockers; ALT, alanine transaminase; AST, aspartate aminotransferase; BMI, body mass index; CCB, calcium channels blockers; GLP-1 RA, glucagon-like peptide-1 receptor agonists; HbA1C, hemoglobin A1C; HDL-C, high-density lipoprotein cholesterol; hsCRP, high-sensitivity C-reactive protein; GFR, glomerular filtration rate; LDL-C, low-density lipoprotein cholesterol; NDO, non-diabetic obese patients; sTSH, supersensitive thyroid stimulating hormone; T2DM, patients with type 2 diabetes mellitus; γ-GTP, gamma-glutamyl transpeptidase. Notes: Data are presented as mean ± SD and analyzed using one-way ANOVA or median (interquartile range) and analyzed using the Kruskal-Wallis H test. Difference of male/female ratio was analyzed using Chi-square test. * indicates *p* < 0.05 between controls vs. NDO. § indicates *p* < 0.05 between controls and T2DM. # indicates a *p* < 0.05 between NDO and T2DM.

**Table 2 ijms-24-04115-t002:** Percentage and absolute amount of lipoprotein subfractions in study participants.

	Controls (n = 49)	NDO (n = 106)	T2DM (n = 62)
VLDL (%)	17.69 ± 3.2	19.9 ± 4.1 *	20.8 ± 5.2 ^§^
IDL (%)	26.6 ± 6.3	25.3 ± 4	24.6 ± 3.9
large LDL (%)	23.2 ± 6.1	28 ± 4.7 *	26.9 ± 5.4 ^§^
small-dense LDL (%)	0.6 (0–1.9)	1.15 (0–2.4)	1.65 (0–3.3) ^§^
mean LDL size (nm)	27.3 (27–27.4)	27.1 (26.9–27.3) *	26.9 (26.9–27.3) ^§,#^
large HDL (%)	29 ± 8.7	22.9 ± 7 *	18.9 ± 6.5 ^§,#^
intermediate HDL (%)	50.3 ± 4.6	51.1 ± 3.7	48.8 ± 4.1
small HDL (%)	20.7 ± 6.3	26 ± 6.8 *	32.2 ± 7.7 ^§,#^
VLDL (mmol/L)	0.89 ± 0.19	1.0 ± 0.24 *	1.1 ± 0.49 ^§^
IDL (mmol/L)	1.34 ± 0.41	1.25 ± 0.30	1.23 ± 0.33
large LDL (mmol/L)	1.17 ± 0.38	1.41 ± 0.36 *	1.33 ± 0.41
small-dense LDL (mmol/L)	0.032 (0–0.093)	0.058 (0–0.13)	0.075 (0–0.185)
large HDL (mmol/L)	0.46 ± 0.26	0.30 ± 0.14 *	0.24 ± 0.12 ^§^
intermediate HDL (mmol/L)	0.73 ± 0.17	0.67 ± 0.17	0.58 ± 0.14 ^§,#^
small HDL (mmol/L)	0.29 ± 0.07	0.34 ± 0.10 *	0.37 ± 0.10 ^§^

Abbreviations: HDL, high-density lipoprotein; IDL, intermediate-density lipoprotein; LDL, low-density lipoprotein; NDO, non-diabetic obese patients; T2DM, patients with type 2 diabetes mellitus; VLDL, very-low density lipoprotein. Notes: Data are presented as mean ± SD and analyzed using one-way ANOVA or median (interquartile range) and analyzed using the Kruskal-Wallis H test. * indicates *p* < 0.05 between controls vs. NDO. § indicates *p* < 0.05 between controls and T2DM. # indicates a *p* < 0.05 between NDO and T2DM.

**Table 3 ijms-24-04115-t003:** Correlations between serum afamin and percentage of high-density lipoprotein (HDL) subfractions in overall subjects (n = 217). HDL subfractions were distributed by Lipoprint gel electrophoresis (Quantimetrix Corp., Redondo Beach, CA, USA) from serum samples.

Afamin vs.	r	*p*
**HDL-1 (%)**	−0.404	<0.001
**HDL-2 (%)**	−0.298	<0.001
**HDL-3 (%)**	−0.462	<0.001
**HDL-4 (%)**	−0.491	<0.001
**HDL-5 (%)**	−0.521	<0.001
**HDL-6 (%)**	0.069	0.3
**HDL-7 (%)**	0.408	0.001
**HDL-8 (%)**	0.221	<0.001
**HDL-9 (%)**	0.391	<0.001
**HDL-10 (%)**	0.483	<0.001
**Large HDL (%)**	−0.47	<0.001
**Intermediate HDL (%)**	−0.28	<0.001
**Small HDL (%)**	0.61	<0.001

## Data Availability

All data generated or analyzed during this study are included in this published article. All the data generated or analyzed during the current study are available from the corresponding author on reasonable request.

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
