# Peer review of "Impaired Organokine Regulation in Non-Diabetic Obese Subjects: Halfway to the Cardiometabolic Danger Zone"

_ijms, 2023, doi:10.3390/ijms24044115_

Round 1

Reviewer 1 Report

I have read with great interest the paper entitled " Impaired organokine regulation in non-diabetic obese subjects: Halfway to cardiometabolic danger zone”. 

The paper is well written and covers a very important issue of cardiometabolic risk in obesity.

I have only minor remarks:

-          The section 4. Materials and methods should be before section 2. Results.

-          Table 1. HbA1c and Insulin – was the difference between NDO and T2DM not significant?

Reviewer 2 Report

In the current study, the authors have reported the expressions (levels) of afamin and its association with several different metabolic players studied in non-diabetic (NDO) obese and type2 diabetic obese (T2DM) subjects. 

The current study showed that Afamin and PAI-1 found to be markedly higher, while RBP4 was lower in the NDO and T2DM group compared to controls.

Afamin was negatively correlated with LDL size and RBP4, whereas negatively correlated with anthropometric, glucose/lipid parameters and PAI-1. Thereby, the BMI, glucose, intermediate HDL and small HDL were predictors of afamin.

The authors propose that organokine regulation in NDO subjects could be predictor of obesity-related comorbidities, where afamin may serve as a biomarker for the severity of cardiometabolic disturbances in obesity. 

Overall, the study is interesting, well designed and the results are presented with clarity. The conclusions are supported with experimental evidence. I have no major comments and endorse the publication of this work.

Notes for the authors:

1.       in the abstract and results or legends it was hard to follow in which sample source these organokine were measured (though mentioned in the methods but abstract and results do not indicate the sample source). please includes the sample type in the abstract and in the results, as well as in the legends. E.g., xxxxx was measured or quantified in the blood samples of….

2.       In figure 1: it would be convenient for readers if authors draw the lines along with p-values, showing the comparison between groups.
